# What the geological past can tell us about the future of the ocean's twilight zone

Katherine A. Crichton [1,6] ✉, Jamie D. Wilson [2,3], Andy Ridgwell [4], Flavia Boscolo-Galazzo [1,7], Eleanor H. John[1], Bridget S. Wade [5] & Paul N. Pearson [1]

Paleontological reconstructions of plankton community structure during warm periods of the Cenozoic (last 66 million years) reveal that deep-dwelling 'twilight zone' (200–1000 m) plankton were less abundant and diverse, and lived much closer to the surface, than in colder, more recent climates. We suggest that this is a consequence of temperature's role in controlling the rate that sinking organic matter is broken down and metabolized by bacteria, a process that occurs faster at warmer temperatures. In a warmer ocean, a smaller fraction of organic matter reaches the ocean interior, affecting food supply and dissolved oxygen availability at depth. Using an Earth system model that has been evaluated against paleo observations, we illustrate how anthropogenic warming may impact future carbon cycling and twilight zone ecology. Our findings suggest that significant changes are already underway, and without strong emissions mitigation, widespread ecological disruption in the twilight zone is likely by 2100, with effects spanning millennia thereafter.

Anthropogenic activities are stressing ocean ecosystems in a multitude of ways, including as a result of pollution, over-fishing, warming, de-oxygenation, increased stratification, and acidification[1]. One of Earth's largest habitats—the mesopelagic ('twilight') zone, which spans ca. 200–1000 m water depth and accounts for approximately one-quarter of the ocean's volume—has so far been somewhat insulated from anthropogenic changes because of the delay involved in propagating perturbations in bulk ocean properties from the surface downwards. Indeed, the twilight zone (hereafter: "TZ") has previously been described as still being "almost pristine"[2]. However, this ignores how other anthropogenic impacts can be propagated nearly instantaneously from the surface to depth in the ocean, meaning that the TZ may already be undergoing significant changes today. Specifically, changes in marine productivity and organic matter export from the ocean surface will be communicated to organisms living in the TZ, which are dependent on sinking particulate organic matter via the "Biological Carbon Pump" (BCP), on time scales of just days to weeks. Furthermore, despite the delay involved in diffusing surface signals

downwards, warming has already started to reach TZ depths[3], hence impacting not only surface ecosystems and carbon export, but the operation of the BCP throughout the upper water column. The potential ecological impact of ocean warming on the delivery of organic matter to subsurface habitats has received little attention.

The BCP may be separated into two broad processes. Firstly, phytoplankton in the photic zone of the upper ocean removes dissolved inorganic carbon (DIC) and nutrients to create new biomass (primary production). Secondly, particulate organic matter (POM) that escapes the ocean surface foodweb gravitationally settles and is delivered to the ocean interior. As POM sinks it is subject to degradation by respiring microbes and other organisms, and progressively transformed back into its dissolved inorganic constituents (including DIC and nutrients) in a process known as 'remineralization'. Integrated across the depth of the water column, remineralization is efficient enough that only about 0.5 to 2%[4] of the original surface export flux reaches the open ocean floor (ca. 2000–6000 m water depth). Ocean mixing and upwelling return DIC and nutrients back to the surface,

[1]School of Earth and Environmental Science, Cardiff University, Cardiff, UK. [2]School of Earth Sciences, University of Bristol, Bristol, UK. [3]Department of Earth, Ocean and Ecological Sciences, University of Liverpool, Liverpool, UK. [4]Department of Earth and Planetary Sciences, University of California, Riverside, CA, USA. [5]Department of Earth Sciences, University College London, London, UK. [6]Present address: Now at Department of Geography, University of Exeter, Exeter, UK. [7]Present address: Now at MARUM, University of Bremen, Bremen, Germany. ✉e-mail: k.a.crichton@exeter.ac.uk

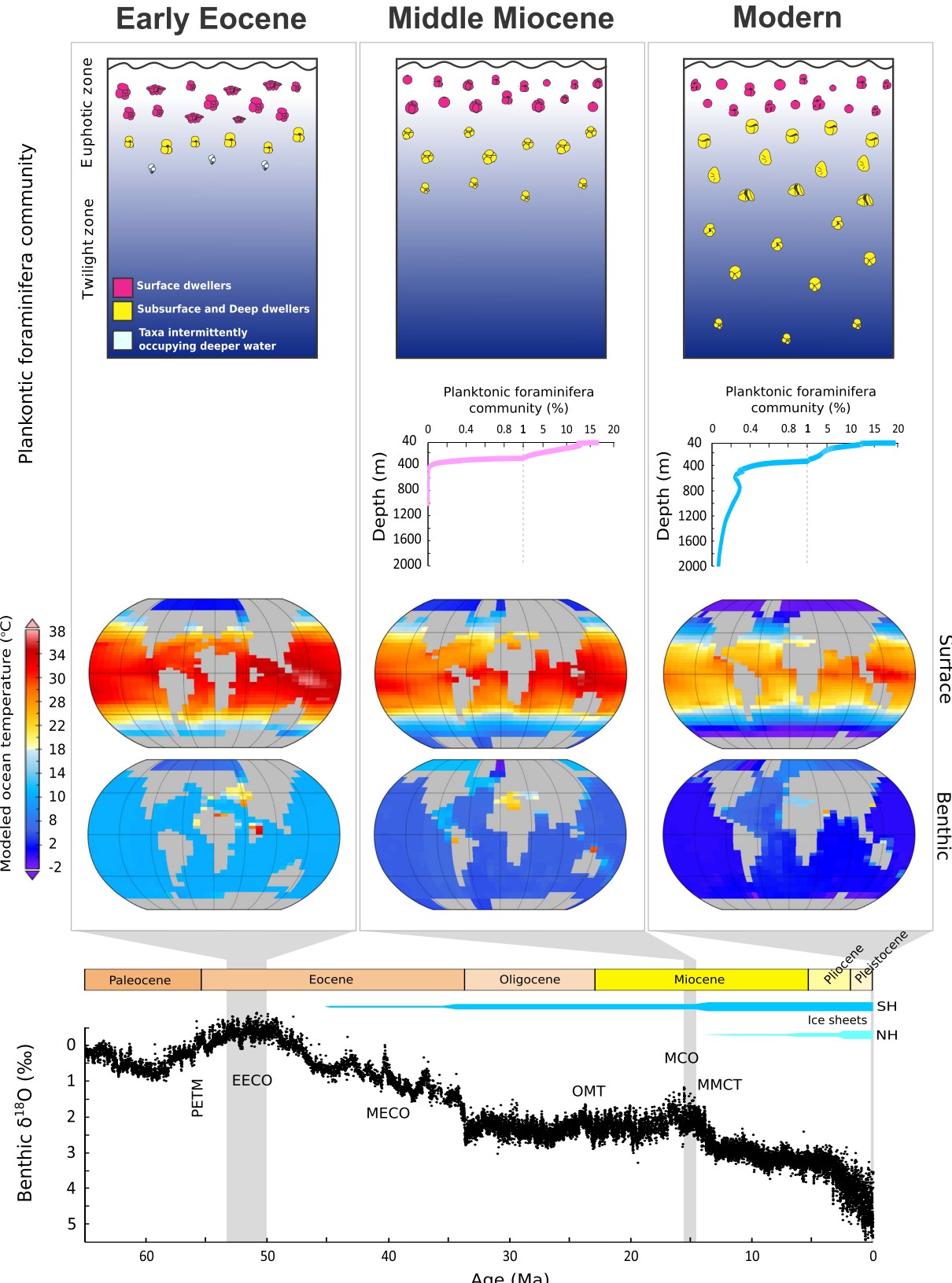

**Fig. 1 | Foraminiferal data and climate indicators for the early Eocene, mid-Miocene, and preindustrial present.** From the top: Illustrations of planktonic foraminiferal distribution and species diversity across the Cenozoic; Planktonic foraminiferal community distribution for the Miocene (pink) and core-top (blue) from ref. 8 (no data available for the early Eocene); modeled surface and benthic ocean temperature and continental configurations for early Eocene, mid-Miocene and preindustrial present ("Modern"); the global benthic δ18O stack for changes over the Cenozoic[79]. The benthic δ18O stack is dominated by the combined effects of temperature and global ice volume (SH is Southern Hemisphere, NH is Northern Hemisphere. PETM is Paleocene Eocene Thermal Maximum, EECO is Early Eocene Climatic Optimum, MECO is Middle Eocene Climatic Optimum, OMT is Oligocene Miocene Transition, MCO is Miocene Climatic Optimum, MMCT is Middle Miocene Climate Transition), as seen in the deep ocean.

completing the cycle. The BCP therefore determines how much carbon is stored in the deep ocean[2] as well as how close to the surface nutrients are released[5], and hence provides feedback on surface productivity, atmospheric $CO_2$, and ultimately, global climate.

Ambient temperature is a master variable that affects the rate of metabolic activity, and hence the operation of the BCP, with each 10 °C increase in temperature resulting in an approximate doubling of metabolic rates[6,7]. As a result, a cool ocean interior helps 'preserve' particles as they sink, facilitating a relatively efficient transfer of organic matter (food for marine biota) from the upper ocean to greater depths[8]. Warming of the water column, in contrast, decreases the efficiency of carbon transfer, reducing food availability at depth. This is critical in respect of the TZ, because this is where the greatest fraction of sinking organic matter is processed by zooplankton and remineralized by microbes[8], and hence where the greatest subsurface transformation of biochemical energy takes place.

The TZ is also a key reservoir of biodiversity and marine biomass. Although total biomass estimates in the TZ are highly uncertain, varying from 1 to 20 billion tonnes[2], even the lower estimates are higher than for epipelagic (to 200 m) and bathypelagic (1000 to 4000 m) waters[9]. Furthermore, the TZ provides a critical temporary habitat for a variety of other organisms, including those that dive in search of prey (e.g., many shark species [10]), and grazers such as lanternfish that "hide" in the twilight zone during the day and migrate to surface waters to feed at night[9,11]. Despite its importance, relatively little is known about the biodiversity and ecology at these depths, and still less about how this complex web may be impacted by a changing supply of food as the ocean warms. The geological past, however, provides us with a route to tackle some of these broad questions.

Although most TZ organisms have a poor or non-existent fossil record, the planktonic foraminifera are an important exception. Different species of these microscopic protists live in depth-stratified habitats from the surface ocean through the thermocline and down into TZ depths. They secrete shells of calcium carbonate that settle through the water column and can accumulate on the seafloor. The chemical and isotopic composition of the shells reveals information about their life environment, and how these changed over thousands to millions of years. Systematic drilling of deep-sea cores over more than fifty years has produced a uniquely valuable archive for investigating the history of the TZ with recovered paleo-records of planktonic foraminifera revealing that both biotic and abiotic factors have steered their evolution over millions of years[12,13]. For instance, the vertical (depth) and horizontal (geographic) distribution of individual species, as well as of the group as a whole, has been subject to major changes linked to long-term climate trends[12,14], transitions[15], and transient warming events[16]. Global numerical modeling of past climates and ocean states (e.g., ref. [17]) provides a means of teasing apart the dominant drivers in these records, as well as their relative contribution to the observed changes. Modeling also provides a means to infer formally, from these past observations, changes that may occur in the future.

In this study, we start by reviewing the fundamental differences in past mesopelagic plankton communities under warmer climatic conditions and show how Earth system modeling of the BCP and associated carbon isotopes has helped explain these patterns. We then build on this, using the same model applied to the past, to investigate what changes may have already occurred since the preindustrial period as well as illustrate what may happen in the future. We focus, in particular, on low-latitude TZ habitats; the high-latitude oceans are more strongly affected by vigorous horizontal and vertical mixing processes that can overprint and obscure the temperature-dependent effects under investigation, as well as generally having a much less complete paleo record. We contrast past, present, and future ocean conditions and critically discuss to what extent observations of past

steady-state conditions can be used in inferring impacts in a dynamically changing present and future world.

## Results

The most recent ~66 million years of Earth's history (the Cenozoic Era) exhibits a rich succession of tectonic, climatic, carbon cycling, and ecological states, summarized in Fig. 1. We focus on two broad periods; the early Eocene, which provides an example of extreme (for the Cenozoic) warmth; and secondly, the overall climate trend from the mid-Miocene through to the present−a 15 million year interval of global cooling which starts with middle Miocene surface ocean temperatures more similar to those we may expect to see in the near future.

### Evidence from the early Eocene

Abundant and diverse proxy evidence indicates that Earth's surface temperatures during the Eocene (~56–33.9 Ma) substantially exceeded modern values[18], with poles that were largely ice-free. Current estimates of atmospheric $CO_2$ concentrations based on the boron isotope ($\delta^{11}B$) proxy range from ~600 to 2500 ppm and suggest that $CO_2$ was the major driver of long-term climate change during the Cenozoic[19,20]. The warmest sustained temperatures were reached during the Early Eocene Climatic Optimum (EECO; ~53−50 Ma) (Fig. 1), with ocean bottom water temperatures of 10 to 12 °C (compared with 2 °C today[21,22]) and global mean sea surface temperatures (SSTs) 10 to 16 °C warmer than the preindustrial[23]. Proxy evidence further suggests that temperatures at mid- to high latitudes were disproportionately elevated compared to the present day, creating reduced meridional[17,24,25] and surface-to-deep temperature gradients as compared to modern[26].

Considering the broader implications of a warmer ocean in the past, Olivarez Lyle and Lyle[27] drew attention to the striking lack of organic carbon present in Eocene sediments, despite independent evidence for high biological productivity at that time. They proposed that elevated ocean temperatures increased the metabolic rates of ocean microbes and, consequently, the rate at which sinking particulate organic carbon (POC) was respired, leaving less to be eventually buried in sedimentary sequences. John et al.[28] provided support for this hypothesis by reconstructing $\delta^{13}C_{DIC}$:depth profiles in the ocean from stable isotope measurements in well-preserved planktonic foraminifera assemblages. These profiles tended to exhibit sharper-than-modern $\delta^{13}C_{DIC}$ gradients in the upper water column, consistent with more efficient recycling of sinking POC and shallower remineralization depths[29].

The early Eocene was also characterized by a pelagic ecosystem that was concentrated in a relatively restricted depth range close to the ocean surface (illustrated in Fig. 1), compatible with the idea that TZ habitats were hostile to respiring organisms because of reduced food supply at depth and upward displacement and intensification of the oxygen minimum zone (OMZ)[28–30]. Indeed, during Eocene transient warming events, barite accumulation rates increased[31], consistent with elevated rates of organic matter remineralization in the water column and the occurrence of food starvation at the seafloor (and hence the coeval low biomass of benthic organisms[31–33]). Conversely, the cooling trend following peak EECO warmth was accompanied by the opening up of new niches for deeper dwelling zooplankton and a consequent increase in biodiversity[34,35], a pattern similar to that also seen earlier during the mid- to late Cretaceous cooling[36,37].

### Evidence from the mid-Miocene to Recent

The most obvious climatic difference between the mid-Miocene and early Eocene is the presence of a continental-scale ice sheet on Antarctica, although neither period shows evidence for a perennially glaciated Greenland. The Miocene Climatic Optimum (MCO) (16.9–14.7 Ma) (Fig. 1) was a time of relative global warmth, when at its peak, mean global ocean temperatures may have reached up to 8 °C

warmer than today[38]. As with the Eocene, elevated warmth was associated with a lower-than-present-day meridional temperature gradient, both on land and in the ocean. At mid and higher latitudes, SSTs in the Southern Hemisphere (30–50°S) and North Sea (56°N) may have reached as high as 30 °C[39,40]. Deep ocean temperatures were also markedly warmer than today, ranging from 8° to 11 °C[22,41–43]—some 6 to 9 °C warmer than present, but still 1 to 6 °C cooler than during the EECO. The MCO was followed by the Middle Miocene Climatic Transition (MMCT) (Fig. 1), a time of step-wise cooling and expansion and stabilization of the Antarctic ice sheet[38]. The MMCT to late Miocene interval is characterized by a major cooling of high and mid-latitude SSTs (around 15 °C)[39,40,43,44], and a more modest cooling of tropical and subtropical latitudes (around 3 °C)[45–47]. Deep ocean temperatures stabilized around 5 °C in the late Miocene, and then underwent further cooling to reach modern-like values (0 to 3 °C) in the early Pliocene (5 Ma)[42].

Deep-sea sediment records of Miocene age are more numerous and widespread than for the early Eocene, allowing for a more complete picture of the operation of the BCP and TZ ecology to be reconstructed. Deep-dwelling (>200 m) planktonic foraminifera were rare in the mid-Miocene, but with a possible shift to deeper habitats at around 13 Ma associated with MMCT cooling[48] and from 8 Ma, an increase in the number of deep living species[49]. For instance, using material from globally distributed sites combined with Earth system modeling, ref. 8 reconstructed planktonic foraminifera species-specific depth habitats from the middle Miocene to the present. They found that planktonic foraminifera species living below 200 m depth were virtually absent in the middle Miocene, with species living

either close to the surface or in the lower part of the euphotic zone (Fig. 1). In-step with cooling from the mid-Miocene onwards, an increase in the number and abundance of species populating the twilight zone occurred, with some species moving from the lower euphotic zone to the TZ, and existing TZ-dwelling species becoming more abundant and moving deeper as global cooling progressed[8]. The late Miocene and Pliocene saw further diversification of both subsurface and deep-dwelling groups, with new species appearing in existing genera, as well as the appearance of new phyletic groups of deep-water taxa[8,34]. Overall, the number of subsurfaces and deep-dwelling planktonic foraminifera species increased from few identified species in the middle Miocene to over a dozen different species by the Holocene, due to a combination of vertical migration and new species origination. This diversification was paralleled by an increase in the depth distribution of planktonic foraminifera (Fig. 1), indicating that from the late Miocene onward, a larger proportion of the community progressively occupied the TZ habitat[8]. Further, ref. 50 showed that the late Miocene to the present evolution of deeper-water forms also involved plankton groups other than foraminifera, such as calcareous nannoplankton.

Along with evidence of an increasingly habitable TZ facilitated by the progressive global cooling occurring since the mid-Miocene, an overall strengthening of the biological carbon pump (and sequestration of more carbon in the deep ocean) can be inferred from a greater difference between surface and benthic δ13C measured in foraminifera tests[51] and a weakening of near-surface δ13C-depth gradients[8]. At the same time, benthic organisms known to form large population blooms when algal aggregates are deposited at the seafloor, increased in global

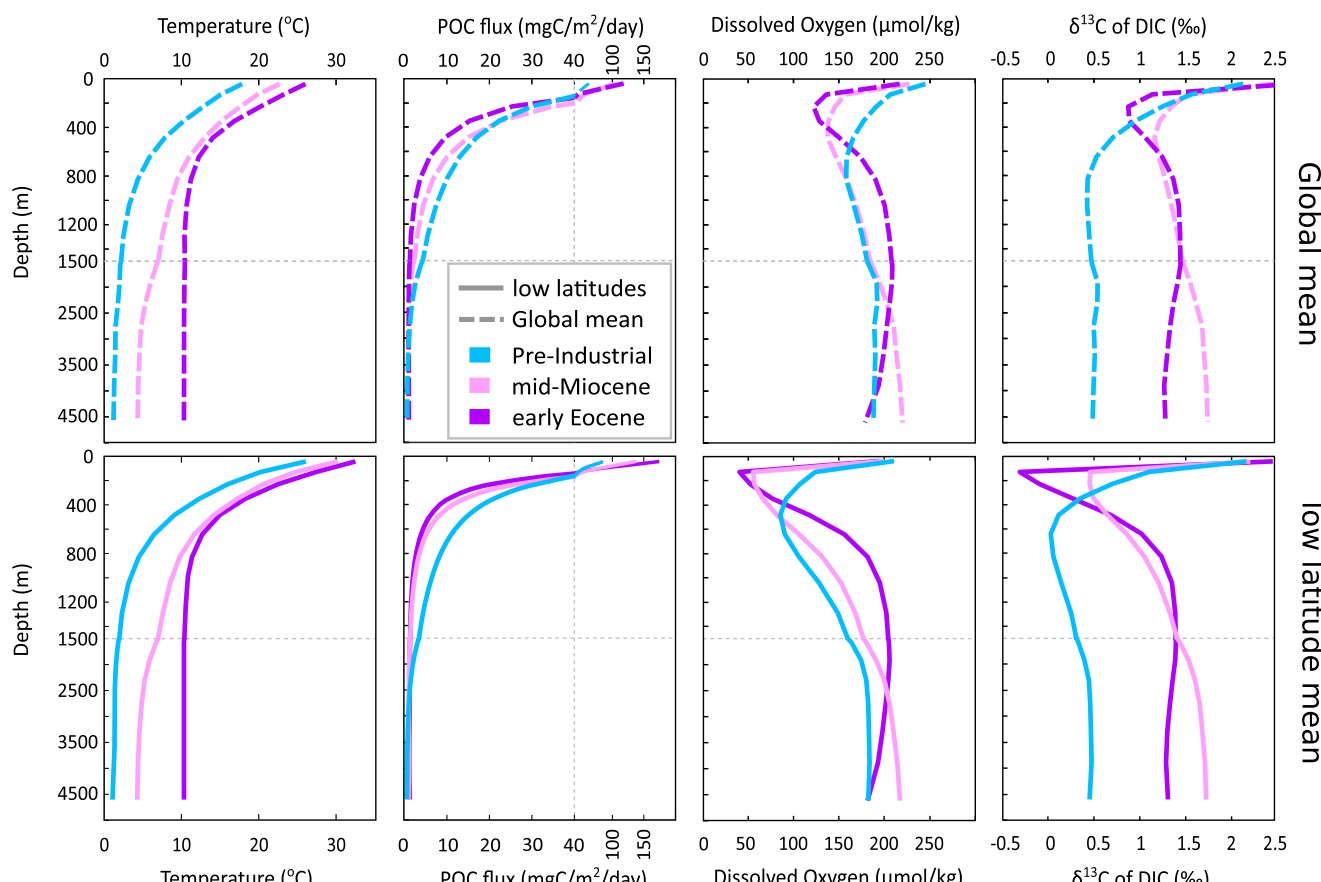

**Fig. 2 | Modeled water column profiles for the early Eocene (purple), mid-Miocene (pink), and preindustrial present (blue).** From left: temperature, particulate organic carbon (POC), dissolved oxygen, and δ13C of DIC (dissolved inorganic carbon) for early Eocene (55 Ma), mid-Miocene (15 Ma), and preindustrial period. Profiles show the modeled global mean properties (dashed lines) and low latitudes mean properties (from 15 °S to 15 °N) (solid lines). Note the change in scale on the depth axes.

abundance[52], further supporting evidence of a more efficient transfer of organic matter to the deep and a more efficient BCP operating in cooler, geologically more recent oceans.

## Evaluating the temperature-dependence of the biological carbon pump of the past

Higher ocean temperatures impact not only the metabolic processes associated with the BCP, but also the cycling and distribution of carbon, nutrients, and dissolved oxygen, which additionally depend on temperature-driven changes in ocean circulation (and gas solubility). Therefore, in assessing the role of temperature on the BCP in the past, we need to account not only for how ocean circulation differs as continental configurations change through time, but also for how circulation responds to different climate forcings. To illustrate how the BCP may have operated differently during the early Eocene and mid-Miocene (matching that in ref. 8), we here create realizations of marine carbon (and attendant oxygen and nutrient) cycling calibrated against both modern and paleo observations[53,54] (and see Methods). We use a version of the "cGENIE" Earth system model in which the BCP includes both a temperature-dependency of nutrient uptakes rates at the ocean surface (and hence organic matter export) as well as of the rate of remineralization of sinking particulate organic matter[53]. It should be noted that while the ocean circulation component of cGENIE is able to reproduce the large-scale structure of the modern ocean, it is low resolution (10° in longitude) and, in the configuration we use here, lacks a coupled dynamical atmosphere. While the model has the advantage of being able to fully equilibrate ocean circulation and the marine carbon cycle for paleo intervals, future projections lack geographic detail and temporal variability and are intended here to be considered as broad illustrative scenarios.

Our modeled early Eocene climate state is characterized by a global mean SST of 27 °C, compared to a little below 24 °C for the modeled mid-Miocene, and 19 °C for the Pre-Industrial period (Fig. 2). The corresponding mean global benthic (deep ocean, defined here as >1000 m water depth) temperatures in the model are 10, 4.6, and 1.5 °C, respectively. Whilst these values approximately align with proxy estimates for the Eocene and Miocene and modern observations, the model struggles to correctly reproduce the less steep meridional temperature gradients under past warmer climates (i.e. the high latitudes remain too cool compared to data, discussed in ref. 54). However, despite the polar ocean surface bias in the model, simulated low-latitude surface and deep ocean temperatures show generally good agreement with paleotemperature data indicators (as shown in ref. 54 where model boundary conditions and set up are described and validated against data). This implies that surface-to-deep temperature gradients are reasonably reproduced in the model across low/mid-latitudes, and hence that the temperature-driven simulated changes in the rate of remineralization with depth through the TZ in this latitudinal band should be plausible.

Ocean warming drives higher metabolic rates associated with bacterial respiration, accelerating the rate and shoaling the depth of organic matter remineralization in the ocean interior, and, in the absence of any change in upper-ocean stratification and hence physical transport, driving a more vigorous recycling of nutrients back to the surface. Increased nutrient supply to the ocean surface can then support higher POC export from the ocean surface, aided by higher surface temperatures and hence higher rates of primary production (photosynthesis) and grazing. The cGENIE model encapsulates this idealized behavior, and while global export is higher during the warmth of the early Eocene, the BCP is inefficient at transferring organic matter from the surface ocean to the deep. For instance, we find that around 2% of POC global export from the surface reaches ~1000 m water depth for the early Eocene, compared to just over 4% for the mid-Miocene, and almost 12% in the preindustrial ocean. In warmer climates, reduced transfer efficiency dominates over the

impact of higher POC export, resulting in an overall weakened/less efficient BCP and lower POC fluxes at depth. In addition, because organic matter degradation occurs faster in the warmer water column and, on average, closer to the surface, the oxygen minimum zone shoals and intensifies upon warming. A comparison of the temperature-dependent cGENIE biological pump model to the standard (non-temperature-dependent BCP) model is available in the supplementary information.

At low latitudes (0° to 15° latitude), where the cGENIE simulations align more closely with proxy temperature estimates and most of the proxy indicators of the BCP are situated, we find that the warmest (early Eocene) model simulation shows the most extreme conditions (Fig. 2): the lowest POC delivery to the TZ depths, the shallowest and most severe oxygen minimum zone, and the sharpest near-surface $\delta^{13}C$ gradients (indicating vigorous nutrient and carbon recycling). The model's $\delta^{13}C_{DIC}$:depth profiles (extracted from the grid cell matching the paleo-location of the data reconstruction) show good agreement with those reconstructed from offshore Tanzania, with shallow $\delta^{13}C$ minima evident in the Eocene water column (see Supplementary Fig. S1 and refs. 28,29). However, only a single Eocene location at which $\delta^{13}C_{DIC}$:depth profiles have been reconstructed are currently available, making the model-reconstructed BCP for the Eocene rather less certain than for the Miocene (and modern). That said, additional observations of widespread low Eocene sedimentary organic carbon content[27,31–33] are consistent with the warm ocean at that time resulting in a pervasively diminished carbon flux to depth.

For the mid-Miocene, the simulated ca. 4 °C warmer-than-modern low-latitude water column temperatures also result in a reduced POC delivery to the TZ and a shoaled oxygen minimum zone as compared to preindustrial (Fig. 2), but less severely than for the early Eocene. Again, modeled mid-Miocene to present $\delta^{13}C_{DIC}$:depth profiles show agreement with those reconstructed from planktonic foraminifer data (see ref. 8 for a full analysis), with shallow $\delta^{13}C$ minima in the mid-Miocene and deepening $\delta^{13}C_{DIC}$ minima as the climate cools (see Supplementary Fig. S1 for the early Eocene, mid-Miocene and core-top preindustrial profiles, and supplementary discussion A). These $\delta^{13}C$-depth model-data comparisons are presented in previously-published work, and the finding of the importance of temperature-dependence on describing characteristics of the past BCP are not new to this study[8,29].

In summary: the geological record shows us that carbon cycling and deep ocean communities were different in warmer oceans, which we can mechanistically explain as a consequence of the temperature-dependence of the BCP[8,28,29]. An important question, then, is whether comparable changes might arise on human time scales as a result of anthropogenic induced warming.

## A temperature-dependent biological pump in the future

The paleoclimate model realizations described above were generated under a constant atmospheric $CO_2$ forcing and therefore represent steady-state patterns of ocean circulation, temperature, and carbon cycling. In contrast, historical and future anthropogenic climate change is characterized by rapid top-down warming from the ocean surface, and a high disequilibrium state of ocean circulation and heat transport. As the BCP strength depends both on surface production (a function, amongst other factors, of the rate of nutrient resupply by ocean circulation) and down-column remineralization and food web processes, an important question is the degree to which changes arising on human time scales as a result of anthropogenically induced transient warming may or may not be similar to past (steady-) states.

To illustrate how future impacts on the TZ may differ from warm intervals in the geological past, we carry out a series of cGENIE model historical-through-future experiments, assuming the same temperature-dependence scheme as that described in 8 and as employed in the paleo analyses. These experiments follow a prescribed

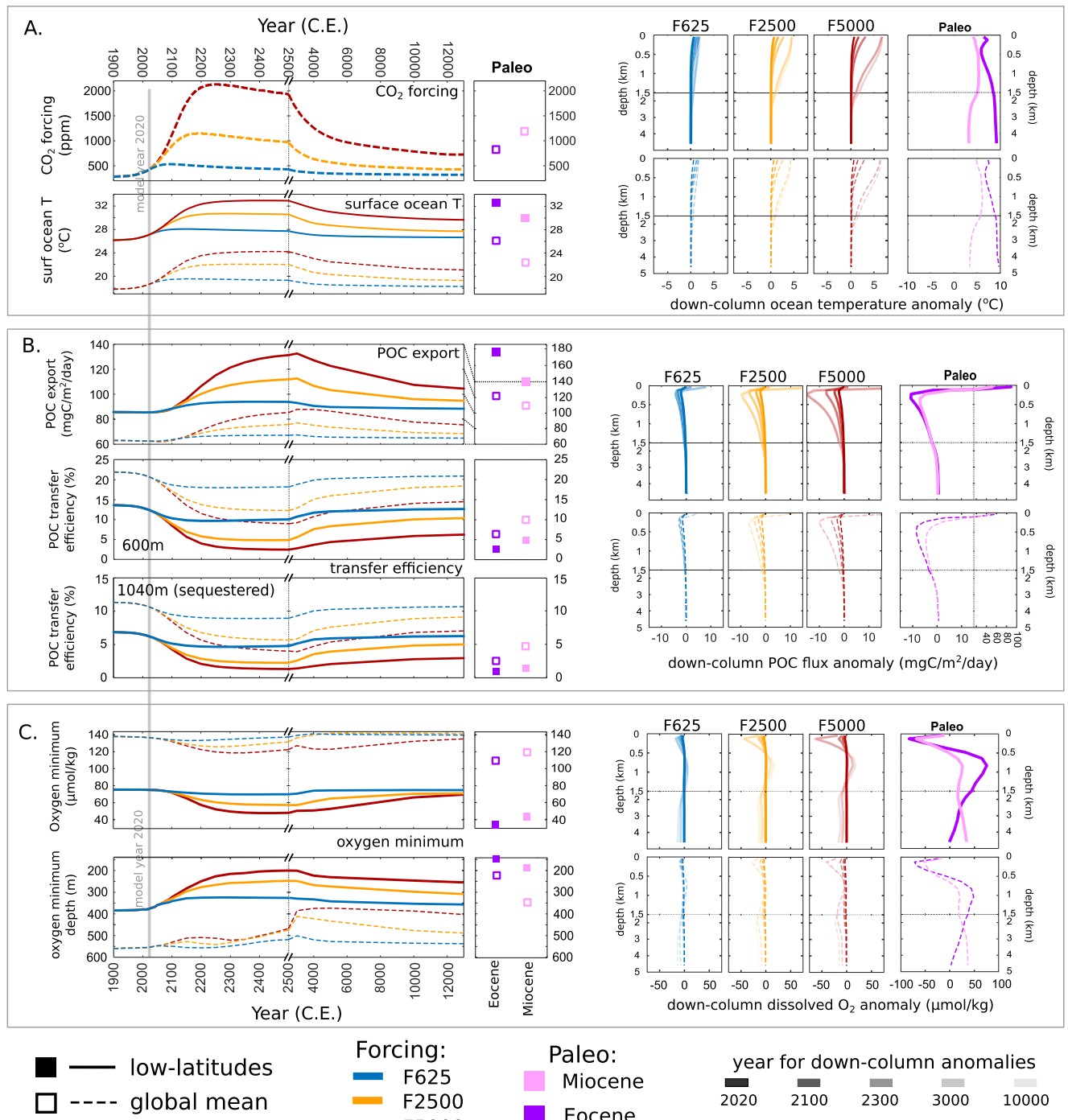

**Fig. 3 | Modeled indicators for projected future scenarios, global (solid lines) and low-latitude (dashed lines) means. A** $CO_2$ forcing, surface ocean temperature, and down-column time slices for temperature as anomalies from the pre-industrial. **B** Particulate organic carbon (POC) exported at 80 m and transfer efficiency to 1040 m (and 600 m the mid-Twilight Zone), and down-column time slices for POC flux as anomalies from the preindustrial (note y-axis scale change between future and paleo). **C** The mean minimum oxygen concentration, the depth of the oxygen minimum, and down-column time slices for dissolved oxygen as anomalies from the preindustrial. Note changes in axes scales for time, depth, and paleo POC flux. Anomalies are all with respect to the preindustrial period. Data for Eocene (purple), Miocene (pink), low emissions F625 (blue), mid emissions F2500 (yellow), and high emissions (red).

trajectory of atmospheric $CO_2$ that includes the historical forcing from 1765 to 2020, followed by three different illustrative future $CO_2$ trajectories (ref. 55, see Methods); one low emission scenario ("F625"), one mid-range ("F2500"), and one high ("F5000"). Each of these scenarios implicitly includes the effect of the neutralization of fossil-fuel $CO_2$ by deep-sea sediments ("carbonate compensation") and terrestrial silicate rock weathering[56], and extend out for several millennia in order to

capture long-term impacts (and to more directly contrast with past climates). It should be noted that the low spatial resolution of the model and lack of inter-annual variability means that the future projections shown here are not detailed and cannot necessarily be considered "robust", particularly with respect to e.g., the CMIP suite of fully coupled Earth system models[57] (we provide a comparison and discussion of the cGENIE model vs. CMIP type models and associated climate

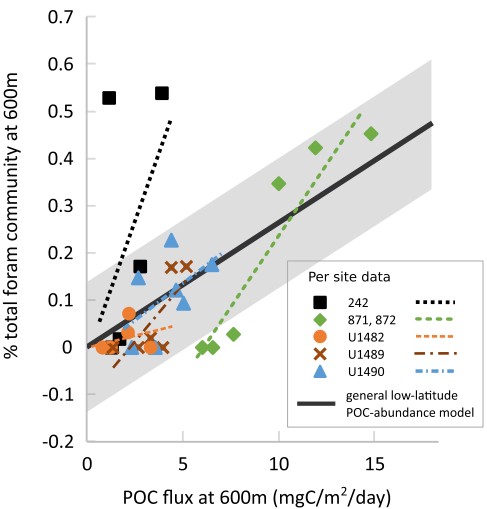

**Fig. 4 | POC-abundance model derived from paleo-abundance data and modeled particulate organic carbon (POC) flux at 600 m.** Data from ref. 8, with general low-latitude linear regression model applied (thick black line). Site-specific linear regressions are shown as dashed lines. The shaded gray is ±1 standard deviation for the general model. Each data point represents data from one site at one time period, covering seven time periods from the mid-Miocene to the pre-industrial present at roughly 2.5 million year time intervals. Data from site 242 (black squares), 871 and 872 (green diamonds), U1482 (orange circles), U1489 (red crosses), and U1490 (blue triangles). More information is available in Methods.

---

change response, in the supplementary discussions B and C). Rather, we seek to directly contrast the past with the future and model with data, all within the same model framework, in order to draw new insights into what the paleo record can and cannot tell us.

Briefly, in response to warming in our idealized future scenarios, the BCP transfer efficiency—here the fraction of POC exported at ~80 m that reaches ~1000 m (the nominal base of the twilight zone)— declines (Fig. 3, note the change in scale on the time axis at the year 2500). This is a direct consequence of the assumed temperature-dependency of the rate of POC remineralization in the model, shoaling the global mean remineralization depth (the depth at which particle flux declines by 63%)[53]. Also in response to warming, POC export increases, reflecting both enhanced nutrient uptake rates with increasing surface temperatures as well as a greater resupply of nutrients to the surface ocean driven by the shoaling of the POC remineralization depth (an effect also seen in other modeling studies[58–60]) modulated by changes in ocean circulation. Similar to past warm climates, the future idealized scenarios result in reduced delivery of POC to depth, despite an increase in POC exported from the surface ocean. However, for future scenarios, the reduction in transfer efficiency occurs sooner than the increase in POC exported from the surface (we find a similar warming impact if we additionally account for a temperature-dependence of DOM cycling, see supplementary discussion D). This lag in POC export with respect to surface temperature arises because subsurface warming lags surface warming, with the delay set by the nutrient return-to-the-surface time scale. This is important because, in general, past warm climates are likely to reflect a quasi steady-state distribution of heat in the ocean and the characteristics of the BCP may then differ with transient future warming.

The Miocene epoch is considered a potential future analog for the climate system, particularly under a 'mid-range' scenario for future carbon emissions[38]. However, despite our simulations showing that future surface ocean temperatures may exceed those of our modeled mid-Miocene (Fig. 3) under mid (F2500) and high (F5000) emissions scenarios by mid 22nd century and early 22nd century, respectively, low-latitude (subsurface) water column temperatures in the model do

not yet resemble those of the modeled Miocene, and only by the 26th century do they start to become comparable. Furthermore, as a result of changes in ocean stratification in the future simulations and the time required for propagation of warming to depth in the model, projected POC export is lower than in the mid-Miocene across all scenarios (Fig. 3). Projected future transfer efficiency to 600 m actually falls below mid-Miocene levels even in the mid-range (F2500) scenario, resulting in POC delivery to mid-TZ depths (ca. 600 m) at low latitudes becoming lower-than-mid-Miocene by mid 23rd century, and lower-than-mid-Miocene POC delivery to the upper twilight zone (200 m) rather earlier, before the end of the current century. So, while future warming may approach mid-Miocene SST values, delayed subsurface warming combined with upper-ocean stratification in response to emissions forcing results in a transient impact on the BCP that qualitatively differs from the impact of steady-state warming. Compared to the mid-Miocene, the cooler subsurface water in future projections results in lower POC export and reduced delivery of POC to the mid-twilight zone in low latitudes.

### Can we use the geological record to inform potential future twilight zone impacts?

Reliable global-scale ocean geochemical observations span only the last ~50 years[61], while marine biological time-series data are even more restricted[62]. Data available from sea-floor derived and sediment trap planktonic community studies indicate that changes have occurred between the preindustrial and present that are consistent with sea surface temperature change[63], suggesting that anthropogenic warming has already altered zooplankton communities. Meta-analyses of marine organisms show evidence of shifting species distributions, altered food webs, variations in community composition, abundance, and distribution in response to thermal change[64,65], with a reduction in ocean productivity over the past century[66]. However, direct evidence of changes occurring in the TZ as a response to warming are lacking, partly because the habitat is difficult to observe and because it has received much less attention[2].

As an alternative to mechanistic forward-modeling of the response of TZ ecology to warming and a weakening BCP, it might be possible to apply a more empirical approach. We illustrate such an approach here based on a compilation of information regarding species diversity and habitat depth of foraminifera (the only group of TZ organisms with an excellent fossil record that existed under different climate and environmental conditions). Specifically, we are interested in the potential for applying past relationships between ecological state and the corresponding environment at the time, to future environmental change. To this end, we take data on how the planktonic foraminifera community distribution with depth has changed with cooling occurring since the mid-Miocene (originally published in ref. 8), to create the simplest possible (linear regression) model linking POC availability with community abundance at mid-TZ depth. Effectively, this could be considered as the inverse of a traditional 'transfer model' that seeks to link modern ecology and environmental conditions, and apply this relationship to past ecology to recover an estimate of the paleoenvironment. In our "POC-abundance" transfer function model, our variables are the model-projected site-specific POC flux at 600 m, and observed site-specific planktonic foraminifer abundance data (as the percentage of total community) at 600 m, across each of seven time slices spanning the mid-Miocene to present (Fig. 4, data from ref. 8). For the linear regression model, we include only sites where a minimum statistical relationship exists between changing POC flux and abundance at 600 m depth over the cooling period, thereby excluding high-latitude sites where physical mixing overprints temperature-dependency as a control on POC availability (see Supplementary Fig. S3 for per-site data and Methods for a complete description of the POC-abundance model).

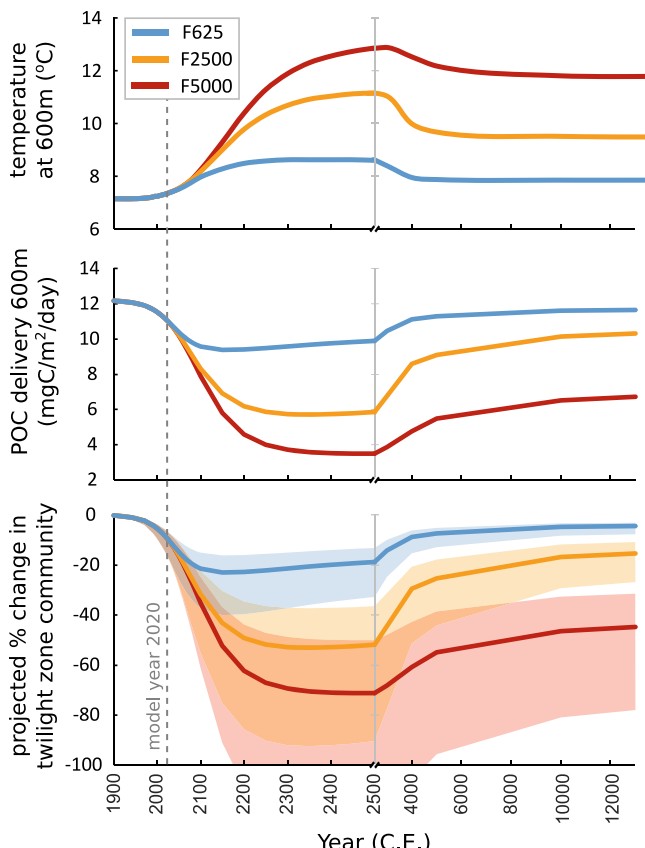

**Fig. 5 | Mean low-latitude temperature and particulate organic carbon (POC) delivery at 600 m depth and projected change in mid-twilight zone community.** Shaded region in the projected % change in the Twilight Zone community is the uncertainty from the POC-abundance model using one standard deviation (Fig. 4). Note x-axis scale change at 2500 C.E. Data for low emissions F625 (blue), mid emissions F2500 (yellow), and high emissions (red).

The datapoints we used to create the POC-abundance model are shown in Fig. 4. We combine all the datapoints (from across all low-latitude sites) to create a general low-latitude linear transfer function model rather than combine site-specific linear regressions (indicated as dashed lines) to avoid site-specific effects skewing the general model. For instance, sites 242, 871, and U1489 appear to show higher sensitivity of planktonic foraminiferal abundance to POC flux than our general model (see supplementary Fig. S3), and only site U1482 shows lower sensitivity. The $R^2$ of the resulting empirical relationship is relatively low at 0.27 (*p* value 0.002). However, we are not seeking to make definitive projections of future TZ ecological impacts, particularly given the caveats to our adoption of a coarse resolution model, but to highlight how data on paleoecological responses to environmental change, mechanistically interpreted with the aid of models, might help shed light on future questions surrounding the biotic impacts of on-going climate change.

For the sake of this exercise, we assume that planktonic foraminifera are representative of the entire TZ ecosystem, on the rationale that food supply is a major driver for all mesopelagic life. Applying the simple POC-abundance model to all three of our future scenarios, we find a drop in abundance in the mid-TZ, in parallel with increasing local water temperature and decreasing POC delivery (Fig. 5). This decrease in abundance is already underway by the model year 2020, and accelerates with the rate of rise in atmospheric $CO_2$ (Figs. 4, 5). Peak impact on the mid-TZ community ranges from a 20% reduction for the low $CO_2$ (F625) scenario to a 70% reduction for the high $CO_2$ (F5000) scenario (central values). Recovery of the twilight zone

ecosystem occurs only gradually over several thousands of years, in parallel to the slow long-term decline of $CO_2$ and climate. Note that in this simple empirical model, abundance is linear and reversible with POC flux, and no extinction can occur.

## Discussion

Our findings suggest that food supply changes will be of critical importance in shaping the future twilight zone habitat. Even for a low emission scenario, food supply to the mid-TZ drops by more than 20% overall, with mid-range and higher-range scenarios involving severe reductions (>50% by the model year 2200). It should be noted, however, that our analysis and empirical POC-abundance model consider only food supply, whereas the paleodata regression may implicitly account for other environmental factors. In particular, a warmer ocean, by increasing metabolic rates, also increases food demand[6,67,68] and oxygen consumption. Oxygen solubility decreases with higher temperature and together with the shoaling and intensification of oxygen minimum zones illustrated here, this would also restrict the availability of suitable animal habitat[50,69] as has recently been suggested in accounting for the observed pattern of end Permian marine extinction[70]. Changing ocean chemistry and acidification adds an additional layer of environmental pressure and complexity, particularly for the calcifying foraminifera inhabiting the TZ[71]. Although we created the empirical model as a function of POC flux, this will co-vary with and hence implicitly reflect to some degree, these other environmental factors.

Future ecosystems may adapt to environmental change, with motile meso- and bathypelagic species potentially being able to migrate through the water column to more suitable depth ranges (e.g., characterized by improved food sources and/or environmental conditions). However, the differing sign of future changes in temperature, oxygen, and food supply may create novel environmental combinations – for example, lower food supply together with lower oxygen— that may challenge the ability of some species to migrate. The past TZ ecosystems we reconstructed would have evolved and be adapted to the prevailing environmental conditions of their time. For instance, the Miocene-through-Holocene colonization of progressively deeper niches in the water column and of increasing diversity[8] reflect an evolutionary sequence that took advantage of the slow emergence of novel environments[50]. In contrast, the very rapid transient nature of future changes may not allow for similar evolutionary novelty, making our empirical POC-abundance model projections only a best-case scenario, and further illustrating how in general, steady-state geological analogs deviate from the transient and disequilibrium nature that can be expected from anthropogenic change.

Our paleodata compilation, together with mechanistic-based modeling, is consistent with the temperature being a key determinant in controlling the distribution of food and oxygen with depth in the ocean, with correlated changes occurring in the species abundance and diversity in the twilight zone. On a broader scale, these links should hold throughout Earth's history. For instance, a recent study[72] concluded that evolving ocean temperature through the Cenozoic, in modulating the flux of organic matter to the ocean floor, was important in accounting for observed trends in the bulk composition and carbon isotopic composition of marine sediments. Further, the temperature-dependent metabolic feedback (rather than the advent of zooplankton or increased phytoplankton size impacting sinking velocity) was found to have dominated global biogeochemical cycling for at least half a billion years[73] and the last billion years when including the impact of oxygenation[74], having major implications for the evolution of the biosphere. A full appreciation of the role of temperature-dependent processes associated with the BCP may then be key, not only for the future, but also to understand the drivers for a variety of evolution and extinction events recorded in the geological record.

## Methods

### Model description

The cGENIE Earth system model consists of a 3D frictional-geostrophic approximation ocean circulation model[75], coupled to a 2D dynamic-thermodynamic sea-ice model[76]. We employ the ocean circulation and sea-ice model on a $36 \times 36$ equal-area grid (10 degrees of longitude and uniform in the sine of latitude), coupled with a 2D energy-moisture-balance atmosphere model[76]. For traceability, we employ a commonly-used configuration of the physical model with 16 vertical levels in the ocean, present-day (for pre-industrial and future simulations) bathymetry, and adopt the physics parameter values and boundary conditions controlling the climate system of ref. 77. The representation of the ocean carbon and other biogeochemical cycles together with ocean-atmosphere gas exchange also follow ref. 77, with the exception of implementing the temperature-dependent configuration of the BCP, described in ref. 53. This temperature-dependent BCP accounts for changes in metabolic rates of nutrient uptake and of POC remineralization with changes in local temperature (but we do not apply a temperature-dependence in dissolved organic carbon processes for those results discussed in the main text). We further, for the Eocene and Miocene configurations, apply paleo boundary conditions, described next.

### Experimental design

We utilized cGENIE paleo-configurations for the Eocene[78] and mid-Miocene[54], combined with concentrations of atmospheric $CO_2$ that produce global ocean temperatures in reasonable agreement with data evidence[54,78]. These are 834 ppm with a $\delta^{13}CO_2$ of −4.5‰ for the Eocene, 1120 ppm and with a $\delta^{13}CO_2$ of −5.3‰ for the mid-Miocene, (and 280 ppm and with a $\delta^{13}CO_2$ of −6.5‰ for the preindustrial). Each model simulation was run for 10,000 years, with the model output discussed in the text taken from the final year of the simulation.

The future projections were run using the preindustrial steady-state simulation as a starting point, to which we then transiently force the model with a prescribed atmospheric $CO_2$ history from Winkelmann et al.[55]. We adopt this particular family of long-term future (+historical) atmospheric $CO_2$ scenarios because they were simulated using a relatively complete representation of the global carbon cycle that includes calcium carbonate sediment-ocean interactions and terrestrial rock weathering. The simulated atmospheric $CO_2$ curves of ref. 55 themselves resulted from a wide range of possible future carbon emissions scenarios that were generated using a logistic function. From these, we took the resulting atmospheric $CO_2$ trajectories corresponding to total $CO_2$ emissions after the year 2010 of 625 PgC (F625) (the low emissions scenario), 2500 PgC (F2500) (mid emissions scenario), and 5000 PgC (F5000) (the high emissions scenario).

### POC-abundance model

We base our empirical model on the planktonic foraminiferal low-latitude abundance data from ref. 8, and directly compare the corresponding modeled POC fluxes at each site and time slice, to derive a generalized relationship linking food availability to the planktonic foraminiferal community distribution at mid-twilight zone depth (600 m). This relationship is shown in Fig. 4 (and supplementary Fig. S2), we apply a simple linear model (Eq. 1) that produces an $R^2$ of 0.27 (and $p$ value of 0.002) for all included data-model points. Each point on the figure represents data from one site at one time slice.

$$\text{Fraction of community}_{600m} = 0.0264 \times \text{POC delivery}(\text{in mgCm}^{-2}\text{day}^{-1})_{600m}$$
(1)

We use the relationship in Eq. 1 to project changes in foraminiferal community distribution for each of our forcing scenarios (Fig. 5).

### Reporting summary

Further information on research design is available in the Nature Portfolio Reporting Summary linked to this article.

## Data availability

All data used is available in the text, in cited work, or in supplementary materials.

## Code availability

The code for the version of the "muffin" release of the cGENIE Earth system model used in this paper is tagged as v0.9.39 and is assigned a https://doi.org/10.5281/zenodo.7708871. Configuration files for the specific experiments presented in the paper can be found in the directory: genie-userconfigs/PUBS/published/Crichton_et_al.2023. Details of the experiments, plus the command line needed to run each one, are given in the readme.txt file in that directory. All other configuration files and boundary conditions are provided as part of the code release. A manual detailing code installation, basic model configuration, tutorials covering various aspects of the model configuration, experimental design, and output, plus the processing of results, is assigned a https://doi.org/10.5281/zenodo.7545814.

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

## Acknowledgements

This research was supported through Natural Environment Research Council (NERC) grant NE/N001621/1 to P.N.P. (F.B.-G. and K.A.C.) and NE/N002598/1 to B.S.W; J.D.W. acknowledges funding from the AXA Research Fund. A.R. acknowledges support via National Science Foundation grants 1658024, 1702913, and EAR-2121165, as well as from the Heising-Simons Foundation. Thanks to Ian Hall for help with guiding the NERC project.

## Author contributions

Conceptualization by P.N.P.; Formal analysis by K.A.C.; Investigation by J.D.W. and K.A.C.; Visualization by K.A.C. and F.B.-G.; Methodology by A.R., P.N.P., and K.A.C.; Funding acquisition by P.N.P., B.S.W., and A.R.; Writing—original draft, review and editing by K.A.C., J.D.W., A.R., P.N.P., F.B.-G., E.H.J., and B.S.W.

## Competing interests

The authors declare no competing interests.
