## [Peer Review File · Nature Communications]

What the geological past can tell us about the future of the ocean's twilight zoneReviewer #1 (Remarks to the Author):

Review of "What can the geological past tell us about the future of the ocean twilight zone?" by K. A. Crichton et al.

Crichton et al. apply a coarse global ocean-climate model to investigate the impact from increased ocean temperatures on the flux of organic matter through the meso-pelagic zone. They consider three time-slices in the Cenozoic; the EECO, MCO and the pre-industrial period, and show that increased temperatures in the past may have decreased the flux of organic carbon through the meso-pelagic. A warmer meso-pelagic zone increases remineralisation of the sinking flux of organic matter and thereby reduces the flux into the bathy-pelagic and to the bottom. They compare model simulations with paleo-data of ^{13}C -gradients in the upper ocean and differences between surface and bottom values and discuss cycling of carbon and nutrients in a warmer climate. Finally they investigate model scenarios with three prescribed future scenarios and develop a linear model of plankton abundance in the meso-pelagic, and simulate future changes in the meso-pelagic plankton community. They conclude that temperature is a key determinant for the distribution of organic carbon in the meso-pelagic zone, and that temperature changes associated with mid-range future emission scenarios will cause significant changes in the meso-pelagic ecosystem.

I find the model simulations of the steady-state conditions during the three periods very interesting and illustrative for understanding the potential impact from warming of the meso-pelagic. The role of increased remineralisation of organic matter in a warmer meso-pelagic is well-known from previous studies, however, this study interestingly investigates this coupling between temperature and carbon cycling in the context of two previous warm periods.

My concerns with the conclusions in this paper is related to three issues that the authors need to address more carefully before I can recommend publication in Nature Communications:

(1) The model is evaluated against paleo-data of ^{13}C and thereby the strength of the biological pump can be assessed during different time-periods and temperature conditions. I acknowledge the limitations involved in this exercise (also discussed in the paper), however, I find that too little attention is paid to this comparison. An example from Eocene is discussed (l.272) off Tanzania where it is argued that there is a good accordance with reconstructed ^{13}C -profiles. Even though the gradients from this location are in good accordance, a discussion is missing of why this sample from the Mozambique Channel is significant for conditions in the global ocean or on a regional scale for the Indian Ocean. The grid-size of the model implies a horizontal scale of ~ 1000 km and therefore a direct comparison requires some consideration of horizontal variability. Comparing data from other locations in the supplementary does not give an impression of a clear temperature signal during the MCO. For example, sample no. 516 from the South Atlantic may represent a larger ocean basin, however, there is no apparent improvement of the MCO-simulation compared with the present-day simulation. Sample no. 1138, located in the Southern Ocean, show no significant difference or improvement when a temperature-dependence is included. These samples are located in major ocean basins but they do not directly support that temperature change plays an important role for the biological carbon pump here. I find that the model evaluation against available ^{13}C from the previous warm periods is central for the validation of the model, and therefore a more detailed and convincing discussion of model performance and figure S1 is needed.

(2) The transient model response to future emission scenarios are discussed in great detail, for example conditions in year 2020 (l. 413), year 2070 (l. 365) and year 2100 (l. 350). I find the transient simulations are speculative and I am not convinced that the model can simulate transient changes in the ocean on decadal or centennial time-scales. I think it needs to be considered more carefully whether the relatively short-term response of the meso-pelagic can be addressed with such a coarse model. There may be previous studies analyzing this issue in the model which I am not aware of, however, further information should be given to support that ocean circulation in the model is

able to respond to transient changes on these short timescales, for example by validating the model against transient tracers.

(3) The plankton-abundance model (Fig. 4) is applied for simulating future changes in the meso-pelagic plankton community. The model has an r^2 of 0.27 and this is quite low and no significance is provided. This is an interesting figure, however, the model is very shortly discussed and I think a r^2 of 0.27 requires further evaluation. There are 2 or 5 outliers which are not discussed. Are all values considered equally important and are the coarse resolution POC-fluxes representative for local conditions at the sample sites?

I also have some general comments that need clarification.

General comments

I. 52: How is the contribution from DOC considered in the model.

I. 250: It is argued that surface warming will drive a higher POC flux out of the surface layer due to increased metabolic rate of photosynthetic plankton. However, photosynthesis is not solely regulating export of organic carbon and empirical studies suggest that primary production only plays a minor role for export production (e.g., Laws et al., Glob. Biogeochem. cycles, 14, 2000). A reference supporting the assumed effect from temperature on export is missing.

Reviewer #2 (Remarks to the Author):

Critchton et al. presented modelled results of twilight zone ecology in the future, in response to primarily, temperature increases. The transfer function used in their model was based on paleoceanographic and micropaleontological data from Earth's Cenozoic history. Their conclusion is that under moderate emission scenarios, twilight zone ecosystem could still be significantly impacted by the temperature-dependent biological pump efficiency. This study addressed an important problem using cutting-edge data and model approaches, which will be a timely contribution to our understanding of ocean ecology and carbon cycle in a changing climate. The manuscript is well written – I enjoyed reading it. I suggest publication with minor modifications.

The authors rightfully made it clear that their focus is examining temperature-dependent response in the mid- to low-latitude ocean, while acknowledging that there are other potential stressors such as acidification, overfishing, pollution etc. Strong vertical and horizontal mixing processes at high latitudes would apparently complicate this relationship and is not a focus of this study. However, I still think physical oceanographic changes, in the past and in the future, should be evaluated a bit more in depth. Even if we ignore the polar oceans, changing physical or chemical stratification of the water column is going to impact the exchange of dissolved (nutrient, O₂, CO₂ etc) and particulate (fecal pellets, POM etc) matter between different water depth, which would have consequences for planktic forams and the biological pump. The authors said "simulated low latitude surface and deep ocean temperatures show agreement with paleotemperature data indicators" – can you provide more details on this? And how does the reconstructed/simulated vertical temperature profile inform us about the evolution of depth/strength of thermocline/pycnocline in mid- to low-latitude? How are these changes going to impact nutrient, O₂, CO₂ or particle settling, and ultimately, the ecosystem of the twilight zone and the biological pump?

A minor issue at L509-510: "These are 834 ppm with a $\delta^{13}\text{C}_{\text{CO}_2}$ of -4.5‰ for the Eocene, 1120ppm and with a $\delta^{13}\text{C}_{\text{CO}_2}$ of -5.3‰ for the mid-Miocene". 1120 ppm of CO₂ for the mid-Miocene seems too high. Or these two CO₂ numbers should be switched? How does your $\delta^{13}\text{C}_{\text{CO}_2}$ data compare with Tipple & Pagani, 2010 Paleoceanography?

Nice work.

Summary of changes made

Substantial additional information has been added to the supplementary material in response to reviewer comments.

- SI A. We include a discussion of the temperature-dependent model (compared to the standard non-temperature dependent model) on the d13C-depth curve at low, mid and high latitudes – so at differing surface-temperature conditions – in order to further illustrate the differences seen in fig S1 between these two model schemes. We also add additional discussion to explain why we have confidence in our site-specific model-data comparison summarised in fig S1 (“Model-data comparison at different sites”), in response to comments from reviewer #1.
- SI B. We focus on ocean circulation in response to comments from reviewer #2, who requested more discussion of this aspect of the study.
- SI C. We include a comparison of cGENIE’s response to transient CO2 forcing to recent results from CMIP model ensembles in response to reviewer #2, that show cGENIE’s response is similar to that seen in many more complex models.
- SI D. We performed future-projections including a temperature-dependent DOM scheme (in addition to the POC scheme) to respond to a specific question from reviewer #1.

We have also made adjustments and additions in the main text to respond to the reviewers comments and to clarify some messages, including but not limited to:

- Included the global-mean profiles of changes in ocean parameters along with low-latitudes in figure 2 and figure 3.
- Added some additional explanation of the POC-abundance model, including the site-specific linear regressions on fig 4.
- A note on the low number of datapoints for the Eocene d13C-depth curve
- A clarification on the POC-abundance model, including the site-specific linear regressions in fig. 4.
- Indicating at the appropriate place in the text the new supporting information available in the supplementary material.

Response to reviewers

We were delighted by the positive reviews. We have incorporated the reviewers suggestions into the revised manuscript, that we feel is now greatly improved. Below we address the reviewers’ feedback

Reviewer #1 (Remarks to the Author):

Review of "What can the geological past tell us about the future of the ocean twilight zone?" by K. A. Crichton et al.

Crichton et al. apply a coarse global ocean-climate model to investigate the impact from increased ocean temperatures on the flux of organic matter through the meso-pelagic zone. They consider three time-slices in the Cenozoic; the EECO, MCO and the pre-industrial period, and show that increased temperatures in the past may have decreased the flux of organic carbon through the meso-pelagic. A warmer meso-pelagic zone increases remineralisation of the sinking flux of organic matter and thereby reduces the flux into the bathy-pelagic and to the bottom. They compare model

simulations with paleo-data of ^{13}C -gradients in the upper ocean and differences between surface and bottom values and discuss cycling of carbon and nutrients in a warmer climate. Finally they investigate model scenarios with three prescribed future scenarios and develop a linear model of plankton abundance in the meso-pelagic, and simulate future changes in the meso-pelagic plankton community. They conclude that temperature is a key determinant for the distribution of organic carbon in the meso-pelagic zone, and that temperature changes associated with mid-range future emission scenarios will cause significant changes in the meso-pelagic ecosystem.

I find the model simulations of the steady-state conditions during the three periods very interesting and illustrative for understanding the potential impact from warming of the meso-pelagic. The role of increased remineralisation of organic matter in a warmer meso-pelagic is well-known from previous studies, however, this study interestingly investigates this coupling between temperature and carbon cycling in the context of two previous warm periods.

We feel that is a fair summary in-line with our intention – the role of increased remineralization of organic matter is known, but its implications have not been fully explored, particularly in the context of the past and how this can inform us about the future. We note that we also go much further than this, and explore a novel environmental-ecological relationship rooted in the role of increased remineralization of organic matter to help extend future ecological projections into the Twilight Zone.

My concerns with the conclusions in this paper is related to three issues that the authors need to address more carefully before I can recommend publication in Nature Communications:

(1) The model is evaluated against paleo-data of ^{13}C and thereby the strength of the biological pump can be assessed during different time-periods and temperature conditions. I acknowledge the limitations involved in this exercise (also discussed in the paper), however, I find that too little attention is paid to this comparison.

All the $d^{13}\text{C}$ for Early Eocene (combining 55 and 49 Ma time-slices), mid Miocene (combining 15 and 14 Ma data), and modern (core-top), and for each Site is shown evaluated against the respective modelled $d^{13}\text{C}$ profiles (for both T-dependent and fixed remineralization profile assumptions) in Figure S1. We have added the modelled surface temperature at each of these locations to the figure S1, and combined with the explanations in SI A, provides a more in-depth discussion of the model-data comparison.

As an aside, we would like to note that all the $d^{13}\text{C}$ data used in this study was previously published elsewhere (John et al 2013, 2014 (28 and 29 in the main text); Boscolo-Galazzo, Crichton et al 2021 (8 in the main text)), and that the assessment of the strength of the biological pump using the Miocene-to-present data was the main subject of the Boscolo-Galazzo, Crichton et al 2021 (8 in main text) study published in the journal Science. It is for this reason that we do not go in to a lot of depth about this model-data comparison in this study, we have tried to make this more clear in the text.

An example from Eocene is discussed (I.272) off Tanzania where it is argued that there is a good accordance with reconstructed ^{13}C -profiles. Even though the gradients from this location are in good accordance, a discussion is missing of why this sample from the Mozambique Channel is significant for conditions in the global ocean or on a regional scale for the Indian Ocean. The grid-size of the model implies a horizontal scale of ~ 1000 km and therefore a direct comparison requires some consideration of horizontal variability.

This dataset is the main existing data for the Eocene that reconstructs the depth- $\delta^{13}\text{C}$ profile that is essential for our model-data comparison although one other site from the Guayabal Formation in Mexico shows a consistent pattern (John et al. 2013, 28 in main text). The quality of preservation of foraminifera tests, required to reconstruct the depth- $\delta^{13}\text{C}$ profile, and the age of the data limits

options. Having said that, the reviewer is mistaken in thinking the Tanzania drill site is in the Mozambique Channel, which has unusual oceanography and deep mixing relating to the confined conditions. In fact, it was quite far to the north in the Eocene, as it is now, and GCMs suggest the site was bathed by a proto Somali current water from the Indian Ocean gyre. Further, the work by Pearson et al. (2007) (30 in main text) states “Analysis of sedimentary facies, seismic data, paleogeography, nannofossil and foraminiferal assemblages” also points to open-ocean conditions at the time of interest (Eocene).

Nevertheless, we do not expect any one location to be representative for conditions in the global ocean, or even in an ocean basin. We have therefore highlighted the fact that the Eocene time-slice depends on a single site (in contrast to the Miocene which is relatively data-rich) and noted it as a potential caveat, in the main text:

“However, although additional observations of for instance, widespread low Eocene sedimentary organic carbon content (27, 31-33) are consistent with the warm ocean at that time resulting in very diminished carbon fluxes to depth, our water column $\delta^{13}\text{C}$ data profile comes from just a single location (Tanzania) in the ocean, making the model-reconstructed BCP for the Eocene rather less certain than for the Miocene (and modern).”

We have also added some explanation in supplementary material section A to note that our sites are not expected to represent entire ocean-basin conditions, and noted modelled surface ocean temperature on fig S1.

Comparing data from other locations in the supplementary does not give an impression of a clear temperature signal during the MCO.

For example, sample no. 516 from the South Atlantic may represent a larger ocean basin, however, there is no apparent improvement of the MCO-simulation compared with the present-day simulation.

In the standard model, the remineralisation curve (i.e. the rate of remineralisation of POC at any particular depth) is prescribed globally, and is tuned to reproduce modern tracers (John et al 2014, 29 main text). The resultant $\delta^{13}\text{C}$ -depth curve is the combined effect of the remineralisation rate, but also of physio-chemical processes like air-sea gas exchange rates and mixing/circulation. In contrast in the temperature-dependent model, remineralisation rates are locally described by the energy available for the respiration process, and are therefore temperature-sensitive. In warmer waters (e.g. low latitudes) in the Tdep model, the effect of higher temperature on remineralisation dominates over the other processes (air-sea gas exchange and mixing/circulation) in controlling near-surface $\delta^{13}\text{C}$. This effect can be seen in the new figure S4 (reproduced below), where in low latitude warm waters the Tdep models shows far steeper $\delta^{13}\text{C}$ -depth gradients than mid or high latitude (cooler surface water) locations. In contrast, the standard model displays smaller differences in $\delta^{13}\text{C}$ -depth gradients between latitudes.

Fig S4.

$\delta^{13}\text{C}$ -depth profiles of the cGENIE model for mean high, mid and low-latitudes for the temperature-dependent cGENIE model (top) and the standard cGENIE model (bottom). The temperature-dependence of metabolic rates in the biological pump have a strong impact on ocean $\delta^{13}\text{C}$ -depth profiles, especially in near-surface waters.

Site 516 is located in mid-latitudes in the South Atlantic, with cooler surface waters than (e.g.) Site 871 (low latitudes), and hence has a less steep near-surface $d^{13}\text{C}$ -depth gradient in Tdep; more similar to the standard models $d^{13}\text{C}$ -depth gradient than at Site 871. Having said this, there is still a difference in the near-surface $d^{13}\text{C}$ -depth gradient for the tdep model at Site 516 compared to the standard model, but the data-points do not give us enough information on which is “better” – but this is why multiple locations were used to test the model against data in Boscolo-Galazzo, Crichton et al. (2021) (8 in main text). The main model-data comparison figure from that paper is provided here to demonstrate that the temperature-dependent model much-better represents the data than the standard model.

Redacted

Figure reproduced from Boscolo-Galazzo, Crichton et al. (2021), where full model-data comparison and description of the datasets applied for the Miocene to Present are available.

To address this in the manuscript: we have added an extended explanation and discussion as supplementary text (section A), together with some additional figures demonstrating the difference temperature-dependence makes on $d^{13}C$ -depth curves at low, mid and high latitudes. We have also added a note to each subplot in fig S1 stating the modelled surface water temperature for each site,

Sample no. 1138, located in the Southern Ocean, show no significant difference or improvement when a temperature-dependence is included. These samples are located in major ocean basins but they do not directly support that temperature change plays an important role for the biological carbon pump here.

The Southern Ocean is strongly affected by vertical mixing and generally lacks the well-stratified thermocline situation seen in open ocean localities in the tropics and subtropics. Hence we would not expect to see a strong signal from temperature dependent remineralization of sinking organic matter. We make this point in the introduction, “the high latitude oceans are more strongly affected by vigorous horizontal and vertical mixing processes that can overprint and obscure the temperature-dependent effects under investigation”.

I find that the model evaluation against available ^{13}C from the previous warm periods is central for the validation of the model, and therefore a more detailed and convincing discussion of model performance and figure S1 is needed.

We have added additional discussion to the supplementary material section A, but would like to note again the model-data comparison is not new to this study, but is the subject of previously published work. John et al. (2014) for the early Eocene (29 in the main text), and Boscolo-Galazzo, Crichton et al. (2021) for the mid-Miocene (8 in the main text).

(2) The transient model response to future emission scenarios are discussed in great detail, for example conditions in year 2020 (l. 413), year 2070 (l. 365) and year 2100 (l. 350). I find the transient simulations are speculative and I am not convinced that the model can simulate transient changes in the ocean on decadal or centennial time-scales. I think it needs to be considered more carefully whether the relatively short-term response of the meso-pelagic can be addressed with such a coarse model. There may be previous studies analyzing this issue in the model which I am not aware of, however, further information should be given to support that ocean circulation in the model is able to respond to transient changes on these short timescales, for example by validating the model against transient tracers.

We appreciate where the reviewer is coming from here. We have since put a great deal of effort into sifting through and mining the CMIP model simulations and making as close as a like-for-like comparison with cGENIE as we can, and find that transient model simulation using cgenie produces results consistent with the more complex models in this context. To illustrate this, we provide a comparison between the cgenie output for near-surface oxygen concentrations with more complex models from the CMIP 6 forced with RCP scenario 8.5 (new figure S8, provided below). The oxygen concentration represents the combined effects of circulation and the biological carbon pump (on oxygen production in surface photosynthesis and oxygen consumption in respiration in subsurface waters). We find patterns in the change in oxygen concentration similar to those in the CMIP6 model-mean.

We also compare the response of Atlantic Meridional Overturning Circulation (AMOC) to fast warming for cGENIE to the CMIP5 models (figure S7 provided below). This comparison is presented in supplementary material (section C) in the revised manuscript, and shows that cGENIEs response is similar to the CMIP5 multi-model means for both near-surface oxygen and for AMOC response.

Overall, we are confident that the model is able to respond to transient changes on these short timescales, at least on a region-scale, in a manner that is consistent (lies within the spread) with the CMIP5 ensemble. We present an extended analysis and discussion on this as part of the SI Section C: C. Future simulations: comparison of cGENIE and CMIP models.

Fig S7. Characteristics of Atlantic overturning circulation for the CMIP5 RCP 8.5 scenario and the cGENIE F5000 simulation in this study. The cGENIE circulation response to CO₂ forcing is similar to the multi-model mean of CMIP5 models.

Fig S8 Change in subsurface oxygen saturation, comparison of CMIP6 models under SSP5-8.5 to the cGENIE F5000 simulation for the years 2080-2099 anomaly with respect to the 1995-2014 baseline (for CMIP6) (CMIP6 figure from Kwiatkowski et al 2020, multi-model mean, with areas of model-agreement represented with black dots).

(3) The plankton-abundance model (Fig. 4) is applied for simulating future changes in the meso-pelagic plankton community. The model has an r^2 of 0.27 and this is quite low and no significance is provided. This is an interesting figure, however, the model is very shortly discussed and I think a r^2 of 0.27 requires further evaluation. There are 2 or 5 outliers which are not discussed. Are all values considered equally important and are the coarse resolution POC-fluxes representative for local conditions at the sample sites?

We agree that there are uncertainties associated with the method we apply, not least the small number of paleo-datapoints available to create the model. In response to the reviewers comment, in figure 4 we have added the site-specific linear regressions to the plot along with the all-datapoint POC-abundance model. The outliers discussed by the reviewer represent site-specific differences for the relationship between POC and abundance. As we are attempting a first estimate to indicate whether temperature may be important for twilight zone response for all low-latitudes to future warming, we have combined all datapoints together to try to reduce any site-specific effects. The relatively low R^2 for the combined data contrasts with some of the site-specific R^2 (see figure S3) which are much higher. We have added this note in the main text “The datapoints we use to create the POC-abundance model are shown in fig. 4 and are all drawn from low-latitude sites. We combine all the datapoints (from across all sites) to create a general low-latitude linear model rather than combine site-specific linear regressions (indicated as dashed lines) to avoid any possible site-specific effects skewing the general model. For instance, Sites 242, 871, and U1489 show higher sensitivity of planktonic foraminiferal abundance to POC flux than our general model (see supplementary figure S3), and only Site U1482 shows lower sensitivity.”

I also have some general comments that need clarification.

General comments

I. 52: How is the contribution from DOC considered in the model.

The DOC is not temperature-dependent in this set-up, however, in response we have added a temperature dependent DOC model to the simulations and discuss the effect it has in supplementary material D. The addition of a temperature dependent DOC decay had a small effect on POC (compared to the effect of temperature dependent POC alone), affecting POC export and transfer efficiency, but the tdep POC processes dominated (see SI D).

I. 250: It is argued that surface warming will drive a higher POC flux out of the surface layer due to increased metabolic rate of photosynthetic plankton. However, photosynthesis is not solely regulating export of organic carbon and empirical studies suggest that primary production only plays a minor role for export production (e.g., Laws et al., Glob. Biogeochem. cycles, 14, 2000). A reference supporting the assumed effect from temperature on export is missing.

On a second look, we realize that we poorly worked and argued this section. We have now turned around the structure of the sentences to focus to be on the consequences of warming in terms of nutrient re-supply (and only later then how warming also facilitates the utilization of increased nutrient supply):

“Ocean warming drives higher metabolic rates associated with bacterial respiration, accelerating the rate and shoaling the depth of organic matter remineralization in the ocean interior, and, in the absence of any change in upper ocean stratification and hence physical transport, driving a more vigorous recycling of nutrients back to the surface. Increased nutrient supply to the ocean surface can

then support higher POC export from the ocean surface, aided by higher surface temperatures and hence rates of primary production and grazing.”

Reviewer #2 (Remarks to the Author):

Crichton et al. presented modelled results of twilight zone ecology in the future, in response to primarily, temperature increases. The transfer function used in their model was based on paleoceanographic and micropaleontological data from Earth’s Cenozoic history. Their conclusion is that under moderate emission scenarios, twilight zone ecosystem could still be significantly impacted by the temperature-dependent biological pump efficiency. This study addressed an important problem using cutting-edge data and model approaches, which will be a timely contribution to our understanding of ocean ecology and carbon cycle in a changing climate.

We thank reviewer 2 for this statement

The manuscript is well written – I enjoyed reading it. I suggest publication with minor modifications.

The authors rightfully made it clear that their focus is examining temperature-dependent response in the mid- to low-latitude ocean, while acknowledging that there are other potential stressors such as acidification, overfishing, pollution etc. Strong vertical and horizontal mixing processes at high latitudes would apparently complicate this relationship and is not a focus of this study. However, I still think physical oceanographic changes, in the past and in the future, should be evaluated a bit more in depth. Even if we ignore the polar oceans, changing physical or chemical stratification of the water column is going to impact the exchange of dissolved (nutrient, O₂, CO₂ etc) and particulate (fecal pellets, POM etc) matter between different water depth, which would have consequences for planktic forams and the biological pump.

The ocean physics and processes not directly related to temperature dependence (tdep) of the biological pump are represented in the standard model simulations. We take account of climate forcing and different continental configurations/bathymetry in our paleo simulations. So, the differences between the standard model and tdep model (for both past and future simulations) are all due to T dependence in the biological pump. We have not discussed these in detail simply to avoid complicating too much the core point of the paper. However, we have added some discussion to the supplementary material:

- *We have added a discussion of the controllers of d13C-depth profiles in the standard and tdep model (supplementary A).*
- *Further, we show the global stream function for the paleo and present simulations and future simulations for AMOC (supplementary B) as further information on how the models circulation is affected by these differing boundary conditions and forcings.*
- *We have added the global mean profiles (along with low-latitudes) to figure 2 and 3 in the main text.*
- *We have also added a comparison of AMOC response to warming in cGENIE and the CMIP5 models together with a comparison of sub-surface oxygen concentrations in response to warming in cGENIE and CMIP6 in supplementary material.*

The authors said “simulated low latitude surface and deep ocean temperatures show agreement with paleotemperature data indicators” – can you provide more details on this?

This statement summarises published work on the cgenie model for the Miocene to present (Crichton et al., 2021, 54 in the main text). In that paper paleo-temperature proxies and benthic d13C data were applied to constrain climate forcing so that surface and deep ocean temperatures were in agreement

with the proxy data. The cited reference in the main text is this paper, and in response to this comment we stress that in the text in the revised manuscript “simulated low latitude surface and deep ocean temperatures show agreement with paleotemperature data indicators (as shown in 54 where model boundary conditions and set up are described and validated against data).”

And how does the reconstructed/simulated vertical temperature profile inform us about the evolution of depth/strength of thermocline/pycnocline in mid- to low-latitude? How are these changes going to impact nutrient, O₂, CO₂ or particle settling, and ultimately, the ecosystem of the twilight zone and the biological pump?

The cgenie model can reproduce present day ocean circulation patterns (Cao et al. 57 in main text) which control the depth of the thermocline. In this sense we rely on the model physics (3D frictional-geostrophic approximation) to inform on what the past thermocline depth would have been, taking into account differing bathymetry, wind forcing, and climate forcing than the present. Considering figure 2, in the warmest period (the early Eocene) the base of the thermocline is more shallow than present, and ocean temperature from about 900m and deeper are quite uniform. In the present the base of the thermocline is deeper. We have not reconstructed the thermocline, but we adjusted climate forcing (as CO₂ and North Atlantic Salinity correction for the Miocene to present) to tune the surface and deep ocean temperatures to proxy data, and then rely on the model circulation to inform us about the thermocline (and pycnocline).

To address the reviewers point, we have added a discussion in the supplementary material (section B) on the impact of the thermocline on our results (i.e. on how we reconstructed the depths of foraminifera in Boscolo-Galazzo, Crichton et al. 2021, 8 in main text, and implications of a different thermocline than that which we model).

We also show the shape of the global-mean and the low-latitude temperature-depth profiles in figure 2 in the main text of the revised version.

A minor issue at L509-510: “These are 834 ppm with a $\delta^{13}\text{C}_{\text{CO}_2}$ of -4.5‰ for the Eocene, 1120ppm and with a $\delta^{13}\text{C}_{\text{CO}_2}$ of -5.3‰ for the mid-Miocene”. 1120 ppm of CO₂ for the mid-Miocene seems too high. Or these two CO₂ numbers should be switched?

This is not a mistake. The mid-Miocene forcing of 1120ppm is described extensively and fully discussed in Crichton et al. (2021) (54 in main text), and should be seen as equivalent CO₂ that also account for differences in climate sensitivity and other greenhouse gases (e.g. methane). Despite this higher forcing, the ocean temperatures are lower than the Eocene.

How does your $\delta^{13}\text{C}_{\text{CO}_2}$ data compare with Tiplle & Pagani, 2010 Paleoceanography?

Our $\delta^{13}\text{C}_{\text{CO}_2}$ values are based only on benthic $\delta^{13}\text{C}$ model-data comparisons from Crichton et al. (2021) (54 in the main text) for the mid-Miocene. For the mid-Miocene, our $\delta^{13}\text{C}_{\text{CO}_2}$ is very close to the upper range of that estimate by Tiplle and Pagani (2010). Crichton et al. (2021) used an inverse-modelling method to identify $\delta^{13}\text{C}_{\text{CO}_2}$ (using ocean $\delta^{13}\text{C}$ data combined with the Miocene boundary conditions in the cGENIE model), differences between these estimates and Tiplle and Pagani (2010) may be due to method differences, and different assumptions in ocean circulation changes.

Nice work. *Thank you!*

Reviewer #1 (Remarks to the Author):

Review of "What can the geological past tell us about the future of the oceans twilight zone?" by Katherine A. Crichton, Jamie D. Wilson, Andy Ridgwell, Flavia Boscolo-Galazzo, Eleanor John, Bridget S. Wade, Paul. N. Pearson.

I find that the paper has improved, and, as I wrote in my previous review, it presents a very interesting synthesis and hypothesis of paleodata and biogeochemical cycling in the mesopelagic during different climatic regimes. Thus, I am generally positive about publication of the paper.

However, I had three main comments in my previous review and the authors have addressed them all. I am still not satisfied with the presentation of the transient response simulations and the plankton abundance model. My reservations are related to the limitation of such a coarse model when applied for short-term simulations. The presented validation figures do not support the model simulation of decadal changes. Also the discussion of the plankton abundance model is not supported by statistical significance levels for the different sites. These two issues need to be considered before I can recommend publication in NC.

In the paragraph starting at l. 320, it is argued that the model can be applied for short-term simulations, e.g., (... l. 326, "The projected responses in physical ocean circulation to anthropogenic CO₂ forcing in cGENIE also show similar characteristics to the multi-model mean response of more complex CMIP5 models, as do projected changes in subsurface oxygen concentration (see supplementary discussions B and C)'), and in the analysis the model response to particular years are discussed (l. 394; year 2070).

Figures in the supplementary are included to support the validation of the transient model simulations, and show the overturning function and a comparison of subsurface O₂. However, dynamics in the upper 600 m can not be validated by these two fields alone, and I am not convinced that the overturning is the most relevant parameter to evaluate. The presentation of the oxygen distribution shows substantial differences between the model and the CMIP6 and these differences are not discussed. I acknowledge that a coarse resolution model include such limitations. However, the transient response simulations are not supported by the shown validation figures, and this should be clear from the paragraph. I would suggest to remove or reformulate the discussion of short-term response from the main text.

More material is included on the formulation of the plankton abundance model in the supplementary. The general trends from each sites are discussed (l. 436, "For instance, Sites 242, 871, and U1489 show higher sensitivity of planktonic foraminiferal abundance to POC flux than our general model (see supplementary Fig. S3), and only Site U1482 shows lower sensitivity."). Significance levels are missing for the correlation coefficients and trends, so it is not possible to evaluate whether the stated higher sensitivity is supported by the data.

In response to comments from reviewer 2 of our revised manuscript, we have made changes to the text with the aim of clarifying the main message of our work:

- Removed the paragraph detailing year-specific changes projected by cGENIE for the future simulations. This is not a core part of the work.
- Simplified the details on model-specific caveats in the main text, again to move the focus away from short-term model projections.
- Edited down the discussion section to refocus on our message of the possible future vulnerability of the mesopelagic zone to future non-steady state fast climate change.

REVIEWER COMMENTS

Reviewer #1 (Remarks to the Author):

Review of "What can the geological past tell us about the future of the oceans twilight zone?" by Katherine A. Crichton, Jamie D. Wilson, Andy Ridgwell, Flavia Boscolo-Galazzo, Eleanor John, Bridget S. Wade, Paul. N. Pearson.

I find that the paper has improved, and, as I wrote in my previous review, it presents a very interesting synthesis and hypothesis of paleodata and biogeochemical cycling in the mesopelagic during different climatic regimes. Thus, I am generally positive about publication of the paper.

We thank the reviewer for noting this essential and most important aspect of our paper, and indeed it is our intention to focus on this, the effect of different climate regimes on the mesopelagic carbon cycle, and impact on Twilight Zone biota.

However, I had three main comments in my previous review and the authors have addressed them all. I am still not satisfied with the presentation of the transient response simulations and the plankton abundance model. My reservations are related to the limitation of such a coarse model when applied for short-term simulations. The presented validation figures do not support the model simulation of decadal changes. Also the discussion of the plankton abundance model is not supported by statistical significance levels for the different sites. These two issues need to be considered before I can recommend publication in NC.

In the paragraph starting at l. 320, it is argued that the model can be applied for short-term simulations, e.g., (... l. 326, "The projected responses in physical ocean circulation to anthropogenic CO₂ forcing in cGENIE also show similar characteristics to the multi-model mean response of more complex CMIP5 models, as do projected changes in sub-surface oxygen concentration (see supplementary discussions B and C)", and in the analysis the model response to particular years are discussed (l. 394; year 2070).

Figures in the supplementary are included to support the validation of the transient model simulations, and show the overturning function and a comparison of subsurface O₂. However, dynamics in the upper 600 m can not be validated by these two fields alone, and I am not convinced that the overturning is the most relevant parameter to evaluate.

The overturning circulation is absolutely essential in determining the distribution of heat between the surface and sub-surface ocean, this is why we selected this parameter to compare with finer resolution models. As the mechanism on which we focus (temperature dependence in the biological

carbon pump) is very dependent on local temperature, ocean heat distribution is a very important factor.

The presentation of the oxygen distribution shows substantial differences between the model and the CMIP6 and these differences are not discussed.

The CMIP5 and 6 model-mean do not clearly demonstrate the large range of responses of these higher resolution models to climate forcing. For example, in Fig S8 panel a) areas of model agreement are marked with black dots. In areas without dots, the higher-resolution models do not agree on the direction or magnitude of the change in sub-surface oxygen levels. In this sense, cGENIE is entirely in-line with CMIP models – showing general trends similar to the CMIP multi-model mean, but some differences – which we cover in the discussion of Fig S8 in supplementary material:

“We find that changes in subsurface oxygen show similar patterns to the CMIP6 multi-model mean for the cGENIE standard model simulation (Fig. S8). The North Pacific reveals strongest O₂ depletion under warming, the northern Indian Ocean shows increases in oxygen concentration. The CMIP6 models have various treatments for temperature dependence, with some models accounting in some way for temperature effects on remineralisation rates and others not (Sefaraian et al., 2020). When including temperature dependence on POM “Tdep (POM)” (and on both POM and DOM remineralisation rates “Tdep (DOM POM)”) sub-surface oxygen depletion occurs in almost all locations, including low latitude locations that in the standard model show an increase in oxygen concentration (for example the northern Indian Ocean, the low-latitude Atlantic, Fig. S8). It is noteworthy that the areas where CMIP6 models show disagreement (areas without black dots in Fig. S5a) are the areas where temperature dependence results in oxygen depletion (rather than oxygen increase in the standard model), these areas are the Indian Ocean, the west Pacific low latitudes, and the low latitude tropical Atlantic. This could account for some of the CMIP6 model disagreement, if some models include temperature dependent remineralisation and some do not.”

I acknowledge that a coarse resolution model include such limitations. However, the transient response simulations are not supported by the shown validation figures, and this should be clear from the paragraph. I would suggest to remove or reformulate the discussion of short-term response from the main text.

We appreciate the point from the reviewer, that coarse models have limitations in terms of projecting short-term changes in climate. Indeed, the projections of cGENIE on decadal timescales are not the message we want to communicate in this study. To this end, we have entirely removed the paragraph detailing the model year specific cGENIE output in the manuscript.

More material is included on the formulation of the plankton abundance model in the supplementary. The general trends from each sites are discussed (l. 436, "For instance, Sites 242, 871, and U1489 show higher sensitivity of planktonic foraminiferal abundance to POC flux than our general model (see supplementary Fig. S3), and only Site U1482 shows lower sensitivity."). Significance levels are missing for the correlation coefficients and trends, so it is not possible to evaluate whether the stated higher sensitivity is supported by the data.

*This “higher sensitivity” is simply referring to the gradient of the linear regression, as a means of illustrating the possible differing site-specific effects that may affect POC-abundance relationships locally. We have slightly adjusted the sentence “For instance, Sites 242, 871, and U1489 **appears** to show higher sensitivity of planktonic foraminiferal abundance to POC flux than our general model...” and added:*

“However, we are not seeking to make definitive projections of future TZ ecological impacts, particularly given the caveats to our adoption of a coarse resolution model, but to highlight how data on paleo ecological responses to environmental change, mechanistically interpreted with the aid of models, might help shed light on future questions surrounding the biotic impacts of on-going climate change.”

We have also added to this revision the R^2 value of the POC-abundance linear model with respect to the datapoints (0.27) noting it is relatively low, and have added also the p-value of the regression (0.002). For the site-specific regressions, there are only very few datapoints available. The low number of datapoints is due to the paucity of data of reconstructions of planktonic foraminifera down-column abundances linked to POC supply.

We thank reviewer 2 again for their review of our revised work.